# Music can be reconstructed from human auditory cortex activity using nonlinear decoding models

Ludovic Bellier [1]*, Anaïs Llorens[1], Déborah Marciano[1], Aysegul Gunduz[2], Gerwin Schalk[3], Peter Brunner[3,4,5], Robert T. Knight[1,6]*

1 Helen Wills Neuroscience Institute, University of California, Berkeley, Berkeley, California, United States of America, 2 J. Crayton Pruitt Family Department of Biomedical Engineering, University of Florida, Gainesville, Florida, United States of America, 3 Department of Neurology, Albany Medical College, Albany, New York, United States of America, 4 Department of Neurosurgery, Washington University School of Medicine, St. Louis, Missouri, United States of America, 5 National Center for Adaptive Neurotechnologies, Albany, New York, United States of America, 6 Department of Psychology, University of California, Berkeley, Berkeley, California, United States of America

* ludovic.bellier@gmail.com (LB); rtknight@berkeley.edu (RTK)

**Data Availability Statement:** All preprocessed data supporting the results of this study are available from the Zenodo repository (DOI: 10.5281/zenodo.7876019).

## Abstract

Music is core to human experience, yet the precise neural dynamics underlying music perception remain unknown. We analyzed a unique intracranial electroencephalography (iEEG) dataset of 29 patients who listened to a *Pink Floyd* song and applied a stimulus reconstruction approach previously used in the speech domain. We successfully reconstructed a recognizable song from direct neural recordings and quantified the impact of different factors on decoding accuracy. Combining encoding and decoding analyses, we found a right-hemisphere dominance for music perception with a primary role of the superior temporal gyrus (STG), evidenced a new STG subregion tuned to musical rhythm, and defined an anterior–posterior STG organization exhibiting sustained and onset responses to musical elements. Our findings show the feasibility of applying predictive modeling on short datasets acquired in single patients, paving the way for adding musical elements to brain–computer interface (BCI) applications.

## Introduction

Music is a universal experience across all ages and cultures and is a core part of our emotional, cognitive, and social lives [1,2]. Understanding the neural substrate supporting music perception, defined here as the processing of musical sounds from acoustics to neural representations to percepts and distinct from music production, is a central goal in auditory neuroscience. The last decades have seen tremendous progress in understanding the neural basis of music perception, with multiple studies assessing the neural correlates of isolated musical elements such as timbre [3,4], pitch [5,6], melody [7,8], harmony [9,10], and rhythm [11,12]. It is now well established that music perception relies on a broad network of subcortical and cortical regions, including primary and secondary auditory cortices, sensorimotor areas, and inferior frontal

**Funding:** This work was supported by the Fondation Pour l'Audition (FPA RD-2015-2 to LB), the National Institutes of Health's National Institute of Biomedical Imaging and Bioengineering (R01-EB026439 and P41-EB018783 to PB), and the National Institutes of Health's National Institute of Neurological Disorders and Stroke (U24-NS109103, U01-NS108916, and R13-NS118932 to PB; R01-NS21135 to RTK). The funders had no role in study design, data collection and analysis, decision to publish, or preparation of the manuscript.

**Competing interests:** The authors have declared that no competing interests exist.

**Abbreviations:** BCI, brain–computer interface; CAR, common average reference; CI, confidence interval; ECoG, electrocorticography; HFA, high-frequency activity; ICA, independent component analysis; iEEG, intracranial electroencephalography; IFG, inferior frontal gyrus; MLP, multilayer perceptron; MNI, Montreal Neurological Institute; MSE, mean squared error; SEM, Standard Error of the Mean; SMC, sensorimotor cortex; STG, superior temporal gyrus; STRF, spectrotemporal receptive field.

gyri (IFG) [13–16]. Despite extensive overlap with the speech perception network [17,18], some brain regions of the temporal and frontal lobes are preferentially activated during music perception [15,19–21]. Recent studies report selective musical activation of different neural populations within the STG and the IFG [22]. Both hemispheres are involved in music processing, with a relative preference for the right hemisphere compared to a left dominance for speech [23,24]. However, an integrated view combining these musical elements and specific brain regions using a single predictive modeling approach applied to a naturalistic and complex auditory stimulus is lacking. In this study, we aimed to specify which brain regions are preferentially engaged in the perception of different acoustic elements composing a song.

Here, we used stimulus reconstruction to investigate the spatiotemporal dynamics underlying music perception. Stimulus reconstruction consists in recording the population neural activity elicited by a stimulus and then evaluating how accurately this stimulus can be reconstructed from neural activity through the use of regression-based decoding models. Reconstructing sensory inputs from recorded neuronal responses is proposed to be "a critical test of our understanding of sensory coding" [25]. What information about the outside world can be extracted from examining the activity elicited in a sensory circuit and which features are represented by different neural populations [26–28]?

We adopted the methodological approach used in speech reconstruction. Music and speech are both complex acoustic signals relying on a multiorder, hierarchical information structure —phonemes, syllables, words, semantics, and syntax for speech; notes, melody, chords, and harmony for music [29]. The idea that music could be reconstructed using the same regression approach as applied to speech is further supported by past studies showing a functional overlap of brain structures involved in speech and music processing [30].

Important advances have been made in reconstructing speech from the neural responses recorded with intracranial electroencephalography (iEEG). iEEG is particularly well suited to study auditory processing due to its high temporal resolution and excellent signal-to-noise ratio [31] and provides direct access to the high-frequency activity (HFA; 70 to 150 Hz), an index of nonoscillatory neural activity reflecting local information processing and linked to single-unit firing [32] and the fMRI BOLD signal [33]. Several studies found that nonlinear models decoding from the auditory and sensorimotor cortices provided the highest decoding accuracy [34,35] and success in reconstructing intelligible speech [36]. This is likely due to their ability to model the nonlinear transformations undergone by the acoustic stimuli in higher auditory areas [37,38].

We obtained a unique iEEG dataset where 29 neurosurgical patients passively listened to the popular rock song *Another Brick in the Wall*, *Part 1* (by *Pink Floyd*), while their neural activity was recorded from a total of 2,668 electrodes directly lying on their cortical surface (electrocorticography (ECoG)). This dataset has been used in previous studies asking different research questions without employing decoding or encoding models [39–43]. Passive listening is particularly suited to our stimulus reconstruction approach, as active tasks involving target detection [3,7,8] or perceptual judgments [6,10], while necessary to study key aspects of auditory cognition, can confound the neural processing of music with decision-making and motor activity adding noise to the reconstruction process. The *Pink Floyd* song used in this dataset constitutes a rich and complex auditory stimulus, able to elicit a distributed neural response including brain regions encoding higher-order musical elements including chords (i.e., at least 3 notes played together), harmony (i.e., the relationship between a system of chords), and rhythm (i.e., the temporal arrangement of notes) [44,45].

We investigated to what extent the auditory spectrogram of the song stimulus could be reconstructed from the elicited HFA using a regression approach. We also quantified the effect of 3 factors on reconstruction accuracy: (1) model type (linear versus nonlinear); (2) electrode

density (the number of electrodes used as inputs in decoding models); and (3) dataset duration to provide both methodological and fundamental insights into the reconstruction process. We then tested whether the reconstructed song could be objectively identified, following a classification-like approach [34]. Given the similar qualities of speech and music and the substantial overlap in their neural substrates, we hypothesized that we would encounter the same limitations as observed in speech reconstruction studies, wherein only nonlinear models provide a recognizable reconstructed stimulus (i.e., a song that a listener could identify, without the vocals being necessarily intelligible), and that decoding accuracy has a logarithmic relationship with both electrode density and dataset duration.

Note that previous studies have applied decoding models to the music domain, employing a classification approach. These studies tested whether decoding models could identify different musical pieces [46] and genres [47,48] or estimate musical attention [49] or expertise level of the listener [50]. A recent study attempted to reconstruct music from EEG data and showed the feasibility of this approach [51]. To our knowledge, we present here the first iEEG study reporting music reconstruction through regression-based decoding models.

In addition to stimulus reconstruction, we also adopted an encoding approach to test whether recent speech findings generalize to music perception. Encoding models predict neural activity at one electrode from a representation of the stimulus. These models have been successfully used to evidence key neural properties of the auditory system [52,53]. In the music domain, encoding models have shown a partial overlap between the neural activity underlying music imagery and music perception [54]. Recent speech studies have found that STG was parcellated along an antero-posterior axis. In response to speech sentences, posterior STG exhibited a transient increase of HFA at the onset of the sentence, while anterior STG exhibited a sustained HFA response throughout the sentence [55,56]. Here, we investigated whether we could observe similar HFA activity profiles, namely onset and sustained, in response to a musical stimulus. Finally, we performed an ablation analysis, a method akin to making virtual lesions [57,58], by removing sets of electrodes from the inputs of decoding models. This method allowed us to assess the importance of anatomical and functional sets of electrodes in terms of how much information they contained about the song stimulus, and if this information is unique or redundant across different components of the music network. We hypothesized that the right STG would have a primary role in representing acoustic information during music perception and that we would observe a similar antero-posterior STG parcellation with sustained and onset responses as observed in the speech domain. Further, we anticipated that other components, tuned to specific musical elements, might emerge and extend this parcellation.

In summary, we used regression-based decoding models to reconstruct the auditory spectrogram of a classic rock song from the neural activity recorded from 2,668 ECoG electrodes implanted in 29 neurosurgical patients, we quantified the impact of 3 factors on decoding accuracy, and we investigated the neural dynamics and regions underlying music perception through the use of encoding models and an ablation analysis.

## Results

### Distribution of song-responsive electrodes

To identify electrodes encoding acoustical information about the song, we fitted spectrotemporal receptive fields (STRFs) for all 2,379 artifact-free electrodes in the dataset, assessing how well the HFA recorded at these sites could be linearly predicted from the song's auditory spectrogram (Fig 1). From a dense, bilateral, predominantly frontotemporal coverage (Fig 2A), we identified 347 electrodes with a significant STRF (Fig 2B; see S1 Fig for a detailed view of each

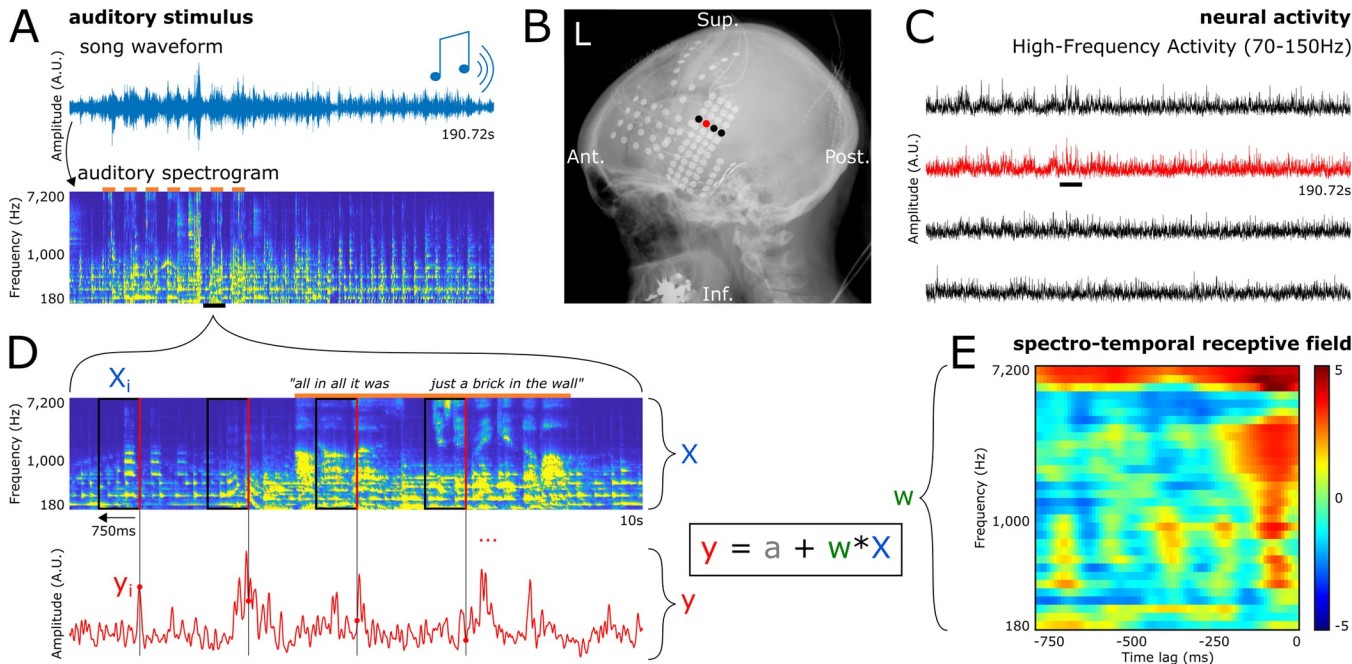

**Fig 1. Protocol, data preparation, and encoding model fitting.** (**A**) Top: Waveform of the entire song stimulus. Participants listened to a 190.72-second rock song (*Another Brick in the Wall, Part 1*, by *Pink Floyd*) using headphones. Bottom: Auditory spectrogram of the song. Orange bars on top represent parts of the song with vocals. (**B**) X-ray showing electrode coverage of 1 representative patient. Each dot is an electrode, and the signal from the 4 highlighted electrodes is shown in (C). (**C**) HFA elicited by the song stimulus in 4 representative electrodes. (**D**) Zoom-in on 10 seconds (black bars in A and C) of the auditory spectrogram and the elicited neural activity in a representative electrode. Each time point of the HFA ($y_i$, red dot) is paired with a preceding 750-ms window of the song spectrogram ($X_i$, black rectangle) ending at this time point (right edge of the rectangle, in red). The set of all pairs ($X_i$, $y_i$), with i ranging from .75 to 190.72 seconds constitute the examples (or observations) used to train and evaluate the linear encoding models. Linear encoding models used here consist in predicting the neural activity (y) from the auditory spectrogram (X), by finding the optimal intercept (a) and coefficients (w). (**E**) STRF for the electrode shown in red in (B), (C), and (D). STRF coefficients are z-valued and are represented as w in the previous equation. Note that 0 ms (timing of the observed HFA) is at the right end of the x-axis, as we predict HFA from the preceding auditory stimulus. The data underlying this figure can be obtained at https://doi.org/10.5281/zenodo.7876019. HFA, high-frequency activity; STRF, spectrotemporal receptive field.

patient's coverage and significant electrodes). We found a higher proportion of song-responsive electrodes in the right hemisphere. There were 199 significant electrodes out of 1,479 total in the left hemisphere and 148 out of 900 in the right one (Fig 2B, 13.5% versus 16.4%, respectively; $X^2$ (1, N = 2,379) = 4.01, p = .045).

The majority of the 347 significant electrodes (87%) were concentrated in 3 regions: 68% in bilateral superior temporal gyri (STG), 14.4% in bilateral sensorimotor cortices (SMCs, on the pre- and postcentral gyri), and 4.6% in bilateral IFG (Fig 2C). The proportion of song-responsive electrodes per region was 55.7% for STG (236 out of 424 electrodes), 11.6% for SMC (45/389), and 7.4% for IFG (17/229). The remaining 13% of significant electrodes were distributed in the supramarginal gyri and other frontal and temporal regions. To examine whether the higher proportion of song-responsive electrodes in the right hemisphere was driven by different nonuniform coverages between both hemispheres (e.g., by a denser coverage of nonauditory regions in the left hemisphere than in the right hemisphere), we restricted our analysis to the 3 main song-responsive regions (STG, SMC, and IFG). We found a higher proportion of song-responsive electrodes in these right song-responsive regions, with 133 significant electrodes out of 374 total, against 165 out of 654 in the corresponding left regions (35.6% versus 25.3%, respectively; $X^2$ (1, N = 1,026) = 12.34, p < .001).

Analysis of STRF prediction accuracies (Pearson's r) found a main effect of laterality (additive two-way ANOVA with laterality and cortical regions as factors; $F(1, 346) = 7.48$,

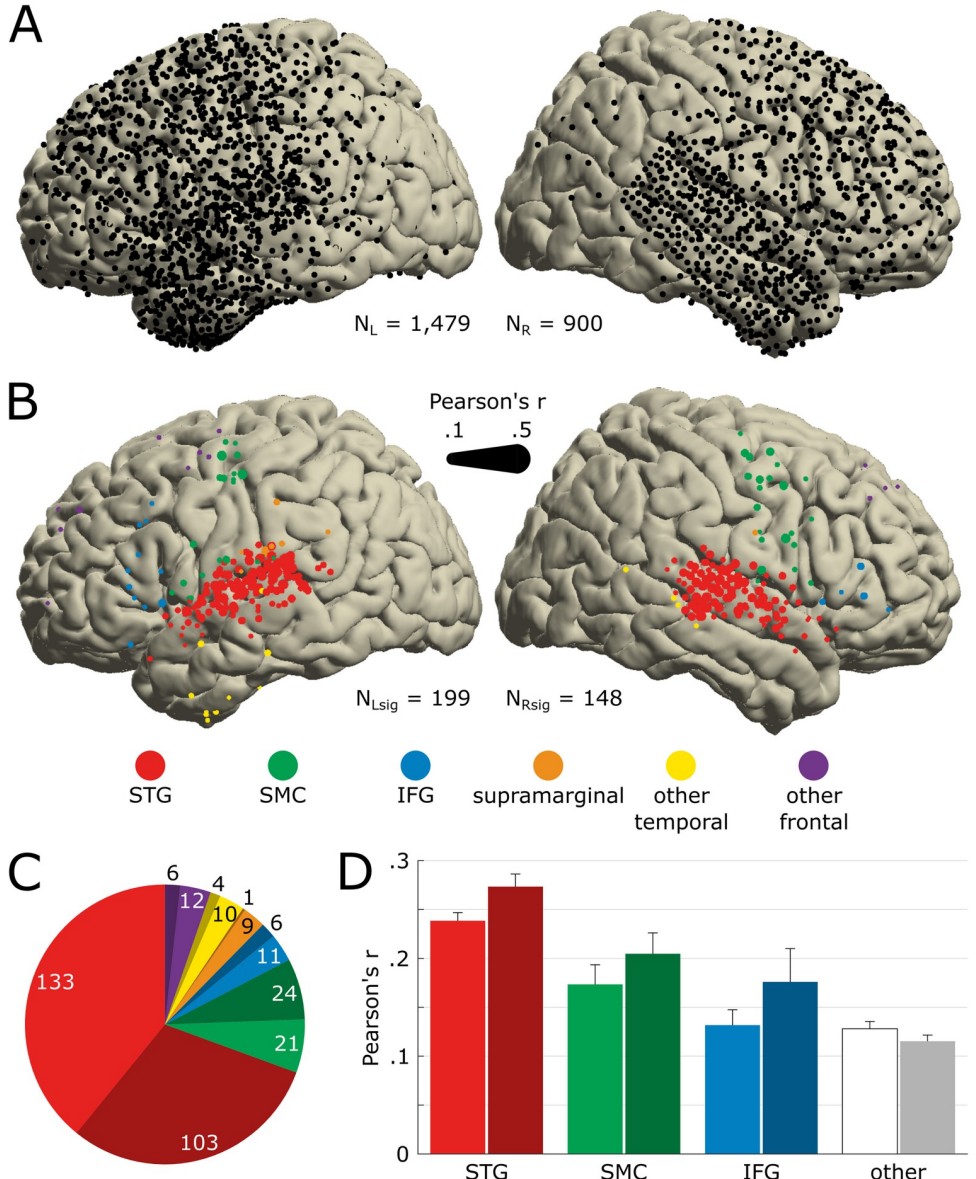

**Fig 2. Anatomical location of song-responsive electrodes.** (**A**) Electrode coverage across all 29 patients shown on the MNI template ($N = 2,379$). All presented electrodes are free of any artifactual or epileptic activity. The left hemisphere is plotted on the left. (**B**) Location of electrodes significantly encoding the song's acoustics ($N_{sig} = 347$). Significance was determined by the STRF prediction accuracy bootstrapped over 250 resamples of the training, validation, and test sets. Marker color indicates the anatomical label as determined using the FreeSurfer atlas, and marker size indicates the STRF's prediction accuracy (Pearson's r between actual and predicted HFA). We use the same color code in the following panels and figures. (**C**) Number of significant electrodes per anatomical region. Darker hue indicates a right-hemisphere location. (**D**) Average STRF prediction accuracy per anatomical region. Electrodes previously labeled as *supramarginal*, *other temporal* (i.e., other than STG), and *other frontal* (i.e., other than SMC or IFG) are pooled together, labeled as *other* and represented in white/gray. Error bars indicate SEM. The data underlying this figure can be obtained at https://doi.org/10.5281/zenodo.7876019. HFA, high-frequency activity; IFG, inferior frontal gyrus; MNI, Montreal Neurological Institute; SEM, Standard Error of the Mean; SMC, sensorimotor cortex; STG, superior temporal gyrus; STRF, spectrotemporal receptive field.

$p = 0.0065$; Fig 2D), with higher correlation coefficients in the right hemisphere than in the left ($M_R = .203$, $SD_R = .012$; $M_L = .17$, $SD_L = .01$). We also found a main effect of cortical regions ($F(3, 346) = 25.09$, $p < .001$), with the highest prediction accuracies in STG (Tukey–Kramer post hoc; $M_{STG} = .266$, $SD_{STG} = .007$; $M_{SMC} = .194$, $SD_{SMC} = .017$, $p_{STGvsSMC} < .001$; $M_{IFG} =$

.154, $SD_{IFG}$ = .027, $p_{STGvsSMC}$ < .001; $M_{other}$ = .131, $SD_{other}$ = .016, $p_{STGvsSMC}$ < .001). In addition, we found higher prediction accuracies in SMC compared to the group not including STG and IFG ($M_{SMC}$ = .194, $SD_{SMC}$ = .017; $M_{other}$ = .131, $SD_{other}$ = .016, $p_{SMCvsOther}$ = .035).

## Song reconstruction and methodological factors impacting decoding accuracy

We tested song reconstruction from neural activity and how methodological factors including number of electrodes included in the model, the dataset duration, or the model type used impacted decoding accuracy. We performed a bootstrap analysis by fitting linear decoding models on subsets of electrodes randomly sampled from all 347 significant electrodes across the 29 patients, regardless of anatomical location. This revealed a logarithmic relationship between how many electrodes were used as predictors in the decoding model and the resulting prediction accuracy (Fig 3A). For example, 80% of the best prediction accuracy (using all 347 significant electrodes) was obtained with 43 (or 12.4%) electrodes. We observed the same relationship at the single-patient level, for models trained on each patient's significant electrodes, although with lower decoding accuracies (solid-colored circles in S2 Fig; for example, 43 electrodes provided 66% of the best prediction accuracy). We observed a similar logarithmic relationship between dataset duration and prediction accuracy using a bootstrap analysis (Fig 3B). For example, 90% of the best performance (using the whole 190.72-second song) was obtained using 69 seconds (or 36.1%) of data.

Regarding model type, linear decoding provided an average decoding accuracy of .325 (median of the 128 models' effective r-squared; IQR .232), while nonlinear decoding using a two-layer, fully connected neural network (multilayer perceptron (MLP)) yielded an average decoding accuracy of .429 (IQR .222). This 32% increase in effective r-squared (+.104 from .325) was significant (two-sided paired *t* test, t(127) = 17.48, *p* < .001). In line with this higher effective r-squared for MLPs, the decoded spectrograms revealed differences between model types, with the nonlinear reconstruction (Fig 3C, bottom) showing finer spectrotemporal details, relative to the linear reconstruction (Fig 3C, middle). Overall, the linear reconstruction (S2 Audio) sounded muffled with strong rhythmic cues on the presence of foreground elements (vocals syllables and lead guitar notes); a sense of spectral structure underlying timbre and pitch of lead guitar and vocals; a sense of harmony (chord progression moving from Dm to F, C, and Dm); but limited sense of the rhythm guitar pattern. The nonlinear reconstruction (S3 Audio) provided a recognizable song, with richer details as compared to the linear reconstruction. Perceptual quality of spectral elements such as pitch and timbre were especially improved, and phoneme identity was perceptible. There was also a stronger sense of harmony and an emergence of the rhythm guitar pattern.

Stimulus reconstruction was also applicable to a single patient with high-density 3-mm electrode spacing coverage. We used nonlinear models to reconstruct the song from the 61 significant electrodes of patient P29 (Fig 3D). These models performed better than the linear reconstruction based on electrodes from all patients (effective r-squared of .363), but decoding accuracy was lower than that obtained with 347 significant electrodes from all patients. On the perceptual side, these single-patient–based models provided a level of spectrotemporal details high enough to recognize the song (S4 Audio). To assess the lower bound of single-patient–based decoding, we reconstructed the song from the neural activity of 3 additional patients (P28, P15, and P16), with fewer electrodes (23, 17, and 10, respectively, as opposed to 61 in P29) and a lower density (1 cm, 6 mm, and 1 cm center-to-center electrode distance, respectively, as opposed to 3 mm in P29), but still covering song-responsive regions (mostly right, left, and left STG, respectively) and with a good linear decoding accuracy (Pearson's r = .387,

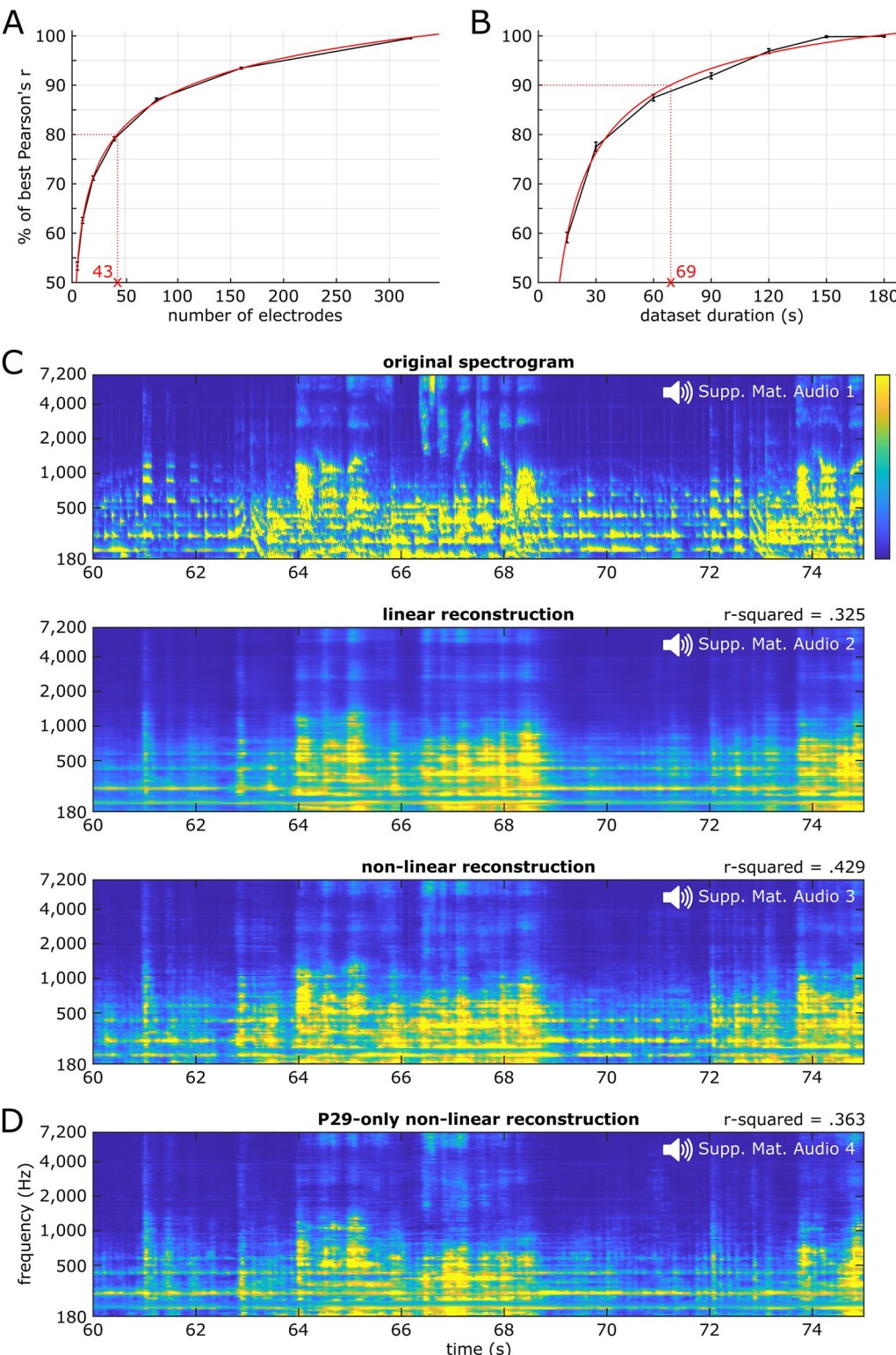

**Fig 3. Song reconstruction and methodological considerations. (A)** Prediction accuracy as a function of the number of electrodes included as predictors in the linear decoding model. On the y-axis, 100% represents the maximum decoding accuracy, obtained using all 347 significant electrodes. The black curve shows data points obtained from a bootstrapping analysis with 100 resamples for each number of electrodes (without replacement), while the red curve shows a two-term power series fit line. Error bars indicate SEM. **(B)** Prediction accuracy as a function of dataset duration. **(C)** Auditory

spectrograms of the original song (top) and of the reconstructed song using either linear (middle) or nonlinear models (bottom) decoding from all responsive electrodes. This 15-second song excerpt was held out during hyperparameter tuning through cross-validation and model fitting and used solely as a test set to evaluate model performance. Corresponding audio waveforms were obtained through an iterative phase-estimation algorithm and can be listened to in S1, S2, and S3 Audio files, respectively. Average effective r-squared across all 128 frequency bins is shown above both decoded spectrograms. **(D)** Auditory spectrogram of the reconstructed song using nonlinear models from electrodes of patient P29 only. Corresponding audio waveform can be listened to in S4 Audio. The data underlying this figure can be obtained at https://doi.org/10.5281/zenodo.7876019. SEM, Standard Error of the Mean.

.322, and .305, respectively, as opposed to .45 in P29). Nonlinear models reconstructed the song spectrogram with an effective r-squared of .207, .257, and .166, respectively (S3 Fig). In the reconstructed waveforms (S5, S6, and S7 Audio files), we retrieved partial vocals (e.g., in P15, "all," "was," and "just a brick" were the only recognizable syllables, as can be seen in the reconstructed spectrogram; S3 Fig, top) and a sense of harmony, although with varying focus in recognizability.

We then quantified the decoded song recognizability by correlating excerpts of the original versus decoded song spectrograms. Both linear (Fig 4A) and nonlinear (Fig 4B) reconstructions provided a high percentage of correct identifications (32/38 and 36/38, respectively; Fig 4, left panels) and significant identification mean percentiles (95.2% and 96.3%, respectively; Fig 4, right panels; 1,000-iteration permutation test, CI95 [.449 .582] for linear, [.447 .583] for nonlinear).

## Encoding of musical elements

We analyzed STRF coefficients for all 347 significant electrodes to assess how different musical elements were encoded in different brain regions. This analysis revealed a variety of spectro-temporal tuning patterns (Fig 5A; see S4 Fig for a detailed view of patient P29's STRFs computed for their 10-by-25, 3-mm-center-to-center grid of electrodes). To fully characterize the relationship between the song spectrogram and the neural activity, we performed an independent component analysis (ICA) on all significant STRFs. We identified 3 components with distinct spectrotemporal tuning patterns, each explaining more than 5% variance and together explaining 52.5% variance (Fig 5B).

The first component (28% explained variance) showed a cluster of positive coefficients (in red, in Fig 5B, top row) spreading over a broad frequency range from about 500 Hz to 7 kHz and over a narrow time window centered around 90 ms before the observed HFA (located at time lag = 0 ms, at the right edge of all STRFs). This temporally transient cluster revealed tuning to sound onsets. This component, referred to as the "onset component," was found exclusively in electrodes located in bilateral posterior STG (Fig 5C, top row, electrodes depicted in red). Fig 6C, top row, showed in red the parts of the song eliciting the highest HFA increase in electrodes possessing this onset component. These parts corresponded to onsets of lead guitar or synthesizer motifs (Fig 6A, blue and purple bars, respectively; see Fig 6E for a zoom-in) played every 2 bars (green bars) and to onsets of syllable nuclei in the vocals (orange bars; see Fig 6D for a zoom-in).

The second component (14.7% explained variance) showed a cluster of positive coefficients (in red, in Fig 5B, middle row) spreading over the entire 750-ms time window and over a narrow frequency range from about 4.8 to 7 kHz. This component, referred to as the "sustained component," was found in electrodes located in bilateral mid- and anterior STG and in bilateral SMC (Fig 5C, middle row). Also, this component correlated best with parts of the song containing vocals, thus suggesting tuning to speech (Fig 6C, middle row, in red; see Fig 6D for a zoom-in).

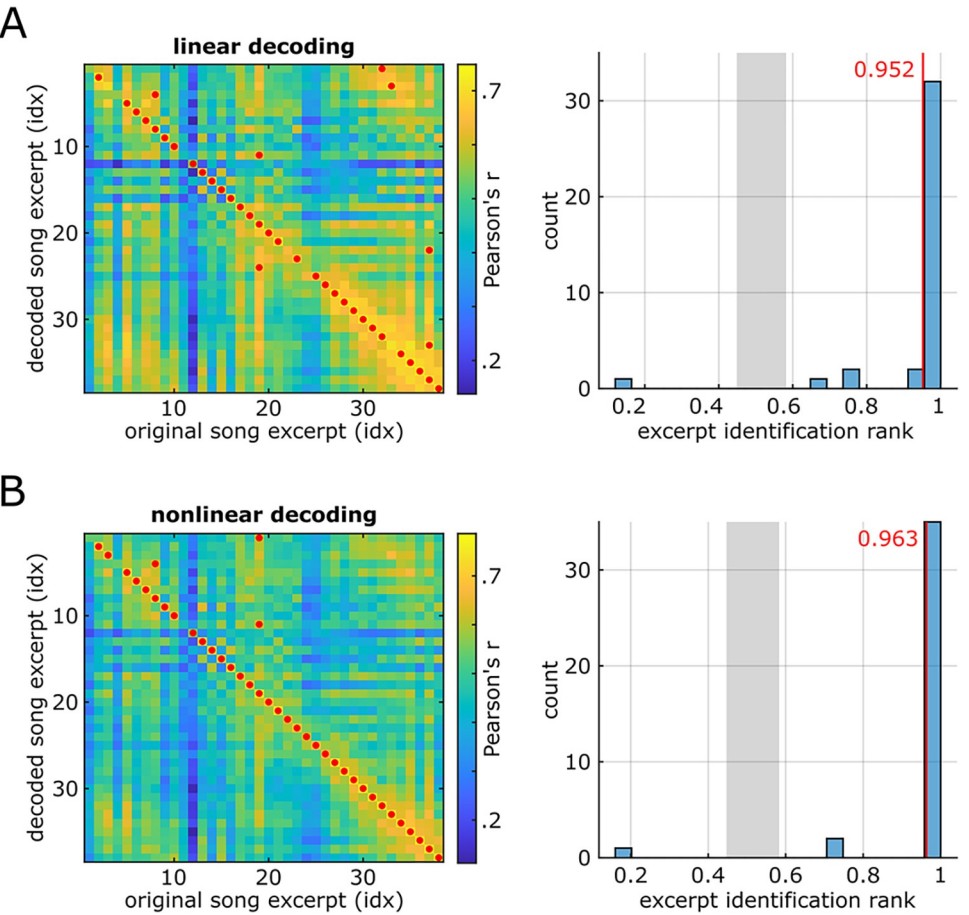

**Fig 4. Song-excerpt identification rank analysis.** After decoding the whole song through 12 distinct 15-second test sets, we divided both the original song and the decoded spectrogram into 5-second excerpts and computed the correlation coefficient for all possible original-decoded pairs. (**A**) Decoding using linear models. Left panel shows the correlation matrix, with red dots indicating the row-wise maximum values (e.g., first decoded 5-second excerpt correlates most with 32nd original song excerpt). Right panel shows a histogram of the excerpt identification rank, a measure of how close the maximum original-decoded correlation coefficient landed from true excerpt identify (e.g., third original-decoded pair correlation coefficient, on the matrix diagonal, was the second highest value on the third excerpt's row, thus ranked 37/38). Gray shaded area represents the 95% confidence interval of the null distribution estimated through 1,000 random permutations of the original song excerpt identities. The red vertical line shows the average identification rank across all song excerpts. (**B**) Same panels for decoding using nonlinear models. The data underlying this figure can be obtained at https://doi.org/10.5281/zenodo.7876019.

The third component (9.8% explained variance) showed a similar tuning pattern as the onset component, only with a longer latency of about 210 ms before the observed HFA (Fig 5B, bottom row). This component, referred to from now on as the "late onset component," was found in bilateral posterior and anterior STG, neighboring the electrodes representing the onset component, and in bilateral SMC (Fig 5C, bottom row). As with the onset component, this late onset component was most correlated with onsets of lead guitar and synthesizer motifs and of syllable nuclei in the vocals, only with a longer latency (Fig 6C, bottom row; see Fig 6D and 6E for zoom-ins).

A fourth component was found by computing the temporal modulations and extracting the maximum coefficient around a rate of 6.66 Hz for all 347 STRFs (Fig 5D, red rectangle). This rate corresponded to the 16th notes of the rhythm guitar, pervasive throughout the song, at the song tempo of 99 bpm (beats per minute). It was translated in the STRFs as small clusters of

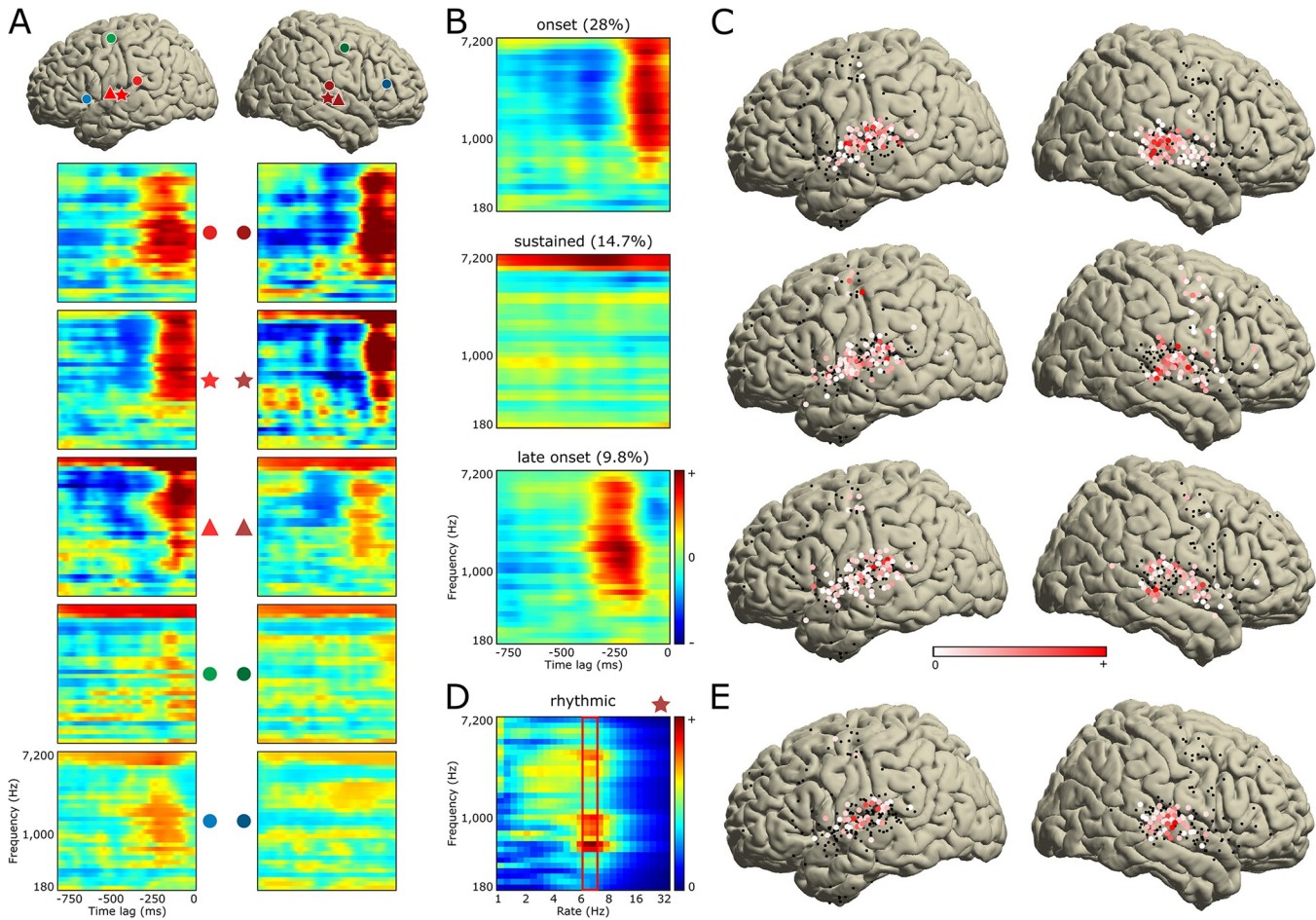

**Fig 5. Analysis of the STRF tuning patterns. (A)** Representative set of 10 STRFs (out of the 347 significant ones) with their respective locations on the MNI template using matching markers. Color code is identical to the one used in Fig 1. **(B)** Three ICA components each explaining more than 5% variance of all 347 significant STRFs. These 3 components show *onset*, *sustained*, and *late onset* activity. Percentages indicate explained variance. **(C)** ICA coefficients of these 3 components, plotted on the MNI template. Color code indicates coefficient amplitude, with in red the electrodes which STRFs represent the components the most. **(D)** To capture tuning to the rhythm guitar pattern (16th notes at 100 bpm, i.e., 6.66 Hz), pervasive throughout the song, we computed temporal modulation spectra of all significant STRFs. Example modulation spectrum is shown for a right STG electrode. For each electrode, we extracted the maximum temporal modulation value across all spectral frequencies around a rate of 6.66 Hz (red rectangle). **(E)** All extracted values are represented on the MNI template. Electrodes in red show tuning to the rhythm guitar pattern. The data underlying this figure can be obtained at https://doi.org/10.5281/zenodo. 7876019. ICA, independent component analysis; MNI, Montreal Neurological Institute; STG, superior temporal gyrus; STRF, spectro-temporal receptive field.

positive coefficients spaced by 150 ms (1/6.66 Hz) from each other (e.g., Fig 5A, electrode 5). This component, referred to as the "rhythmic component," was found in electrodes located in bilateral mid-STG (Fig 5E).

## Anatomo-functional distribution of the song's acoustic information

To assess the role of these different cortical regions and functional components in representing musical features, we performed an ablation analysis using linear decoding models. We first computed linear decoding models for each of the 32 frequency bins of the song spectrogram, using the HFA of all 347 significant electrodes as predictors. This yielded an average prediction accuracy of .62 (Pearson's r; min .27—max .81). We then removed (or *ablated*) anatomically or functionally defined sets of electrodes and computed a new series of decoding models to assess how each ablation would impact the decoding accuracy. We used prediction accuracies

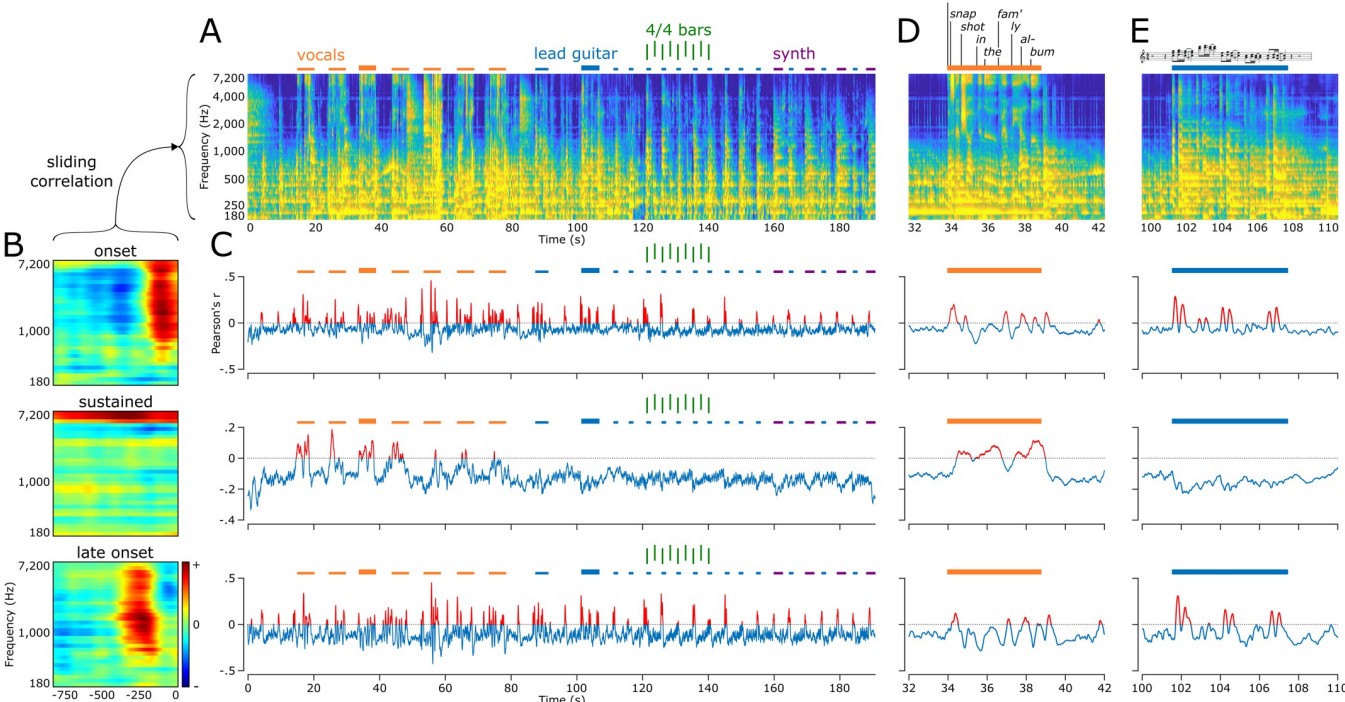

**Fig 6. Encoding of musical elements.** (**A**) Auditory spectrogram of the whole song. Orange bars above the spectrogram mark all parts with vocals. Blue bars mark lead guitar motifs, and purple bars mark synthesizer motifs. Green vertical bars delineate a series of eight 4/4 bars (or measures). Thicker orange and blue bars mark locations of the zoom-ins presented in (D) and (E), respectively. (**B**) Three STRF components as presented in Fig 5B, namely onset (top), sustained (middle), and late onset (bottom). (**C**) Output of the sliding correlation between the song spectrogram (A) and each of the 3 STRF components (B). Positive Pearson's r values are plotted in red, marking parts of the song that elicited an increase of HFA in electrodes exhibiting the given component. Note that for the sustained plot (middle), positive correlation coefficients are specifically observed during vocals. Also, note for both the onset and late onset plots (top and bottom, respectively), positive r values in the second half of the song correspond to lead guitar and synthesizer motifs, occurring every other 4/4 bar. (**D**) Zoom-in on the third vocals. Lyrics are presented above the spectrogram, decomposed into syllables. Most syllables triggered an HFA increase in both onset and late onset plots (top and bottom, respectively), while a sustained increase of HFA was observed during the entire vocals (middle). (**E**) Zoom-in on a lead guitar motif. Sheet music is presented above the spectrogram. Most notes triggered an HFA increase in both onset and late onset plots (top and bottom, respectively), while there was no HFA increase for the sustained component (middle). The data underlying this figure can be obtained at https://doi.org/10.5281/zenodo.7876019. HFA, high-frequency activity; STRF, spectrotemporal receptive field.

of the full, 347-electrode models as baseline values (Fig 7). We found a significant main effect of electrode sets (one-way ANOVA; $F[1, 24] = 78.4$, $p < .001$). We then ran a series of post hoc analyses to examine the impact of each set on prediction accuracy.

**Anatomical ablations (Fig 7A).** Removing all STG or all right STG electrodes impacted prediction accuracy ($p < .001$), with removal of all STG electrodes having the highest impact compared to all other electrode sets ($p < .001$). Removal of right STG electrodes had higher impact than left STG removal ($p < .001$), and no impact of removing left STG electrodes was found ($p = .156$). Together, this suggests that (1) bilateral STG represented unique musical information compared to other regions; (2) right STG had unique information compared to left STG; and (3) part of the musical information in left STG was redundantly encoded in right STG. Ablating SMC, IFG, or all other regions did not impact prediction accuracy ($p > .998$). Removing either all left or all right electrodes significantly reduced the prediction accuracy ($p < .001$), with no significant difference between all left and all right ablations ($p = 1$). These results suggest that both hemispheres represent unique information and contribute to song decoding. Furthermore, the fact that removing single regions in the left hemisphere had no impact but removing all left electrodes did suggests redundancy within the left hemisphere, with musical information being spatially distributed across left hemisphere regions.

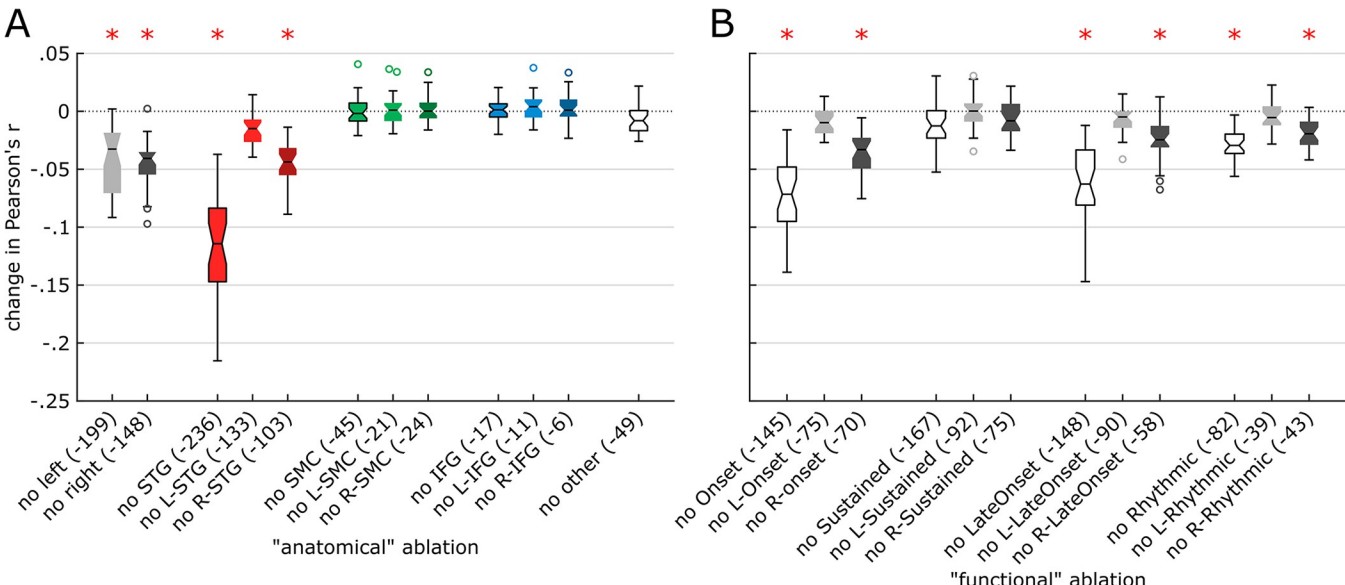

**Fig 7. Ablation analysis on linear decoding models.** We performed "virtual lesions" in the predictors of decoding models, by ablating either anatomical (**A**) or functional (**B**) sets of electrodes. Ablated sets are shown on the x-axis, and their impacts on the prediction accuracy (Pearson's r) of linear decoding models, as compared to the performance of a baseline decoding model using all 347 significant electrodes, are shown on the y-axis. For each ablation, a notched box plot represents the distribution of the changes in decoding accuracy for all 32 decoding models (one model per frequency bin of the auditory spectrogram). For each box, the central mark indicates the median; the notch delineates the 95% confidence interval of the median; bottom and top box edges indicate the 25th and 75th percentiles, respectively; whiskers delineate the range of nonoutlier values; and circles indicate outliers. Red asterisks indicate significant impact from ablating a given set of electrodes. The data underlying this figure can be obtained at https://doi.org/10.5281/zenodo.7876019.

**Functional ablations (Fig 7B).** Removing all onset electrodes and right onset electrodes both impacted prediction accuracy ($p < .001$), with a highest impact for all onset ($p < .001$). No impact of removing left onset electrodes was found ($p = .994$). This suggests that right onset electrodes had unique information compared to left onset electrodes and that part of the musical information in left onset electrodes was redundantly encoded in right onset electrodes. A similar pattern of higher right hemisphere involvement was observed with the late onset component ($p < .001$). Removing all rhythmic and right rhythmic electrodes both significantly impacted the decoding accuracy ($p < .001$ and $p = .007$, respectively), while we found no impact of removing left rhythmic electrodes ($p = 1$). We found no difference between removing all rhythmic and right rhythmic electrodes ($p = .973$). This suggests that right rhythmic electrodes had unique information, none of which was redundantly encoded in left rhythmic electrodes. Despite the substantial number of sustained electrodes, no impact of removing any set was found ($p > .745$). Note that as opposed to anatomical sets, functional sets of electrodes partially overlapped. This impeded our ability to reach conclusions regarding the uniqueness or redundancy of information between functional sets.

## Discussion

We applied predictive modeling analyses to iEEG data obtained from patients listening to a *Pink Floyd* song. We were able to reconstruct the song from direct human neural recordings with the most robust effects using nonlinear models. Through an integrative anatomo-functional approach based on both encoding and decoding models, we confirmed a right-hemisphere preference and a primary role of the STG in music perception, evidenced a new STG subregion tuned to musical rhythm, and defined an anterior–posterior STG organization

exhibiting sustained and onset responses to musical elements. Together, these results further our understanding of the neural dynamics underlying music perception.

Compared with linear models, nonlinear models provided the highest decoding accuracy (r-squared of 42.9%), a more detailed decoded spectrogram, a recognizable song, and a higher rate of song-excerpt identification. This shows that previous methodological findings in speech decoding [34,35] also apply to music decoding. In contrast, linear models had lower decoding accuracy (r-squared of 32.5%) and yielded a smoother decoded spectrogram lacking fine details. This is likely due to the fact that acoustic information represented at STG is nonlinearly transformed, thus requiring the use of nonlinear models to best analyze the electrophysiological data [37]. Note that the overall correlation coefficient computed across all 29 participants for linear decoding models in our present study (r = .29, from the nonnormalized values of S2 Fig) is comparable to the overall decoding accuracy of linear models for pure speech stimuli (r = .28, across 15 participants) [34], suggesting that this limitation is shared between speech and music. Still, linear models yielded strong performance in our classification-like approach, suggesting they could constitute a strategy for some brain–computer interface (BCI) applications, given they are faster to train and easier to interpret than nonlinear models.

We quantified the impact of the number of electrodes used as inputs for the decoding models on their prediction accuracy and found that adding electrodes beyond a certain amount had diminishing returns, in line with previous literature for speech stimuli [34,35]. Decoding accuracy was also impacted by the functional and anatomical features of the electrodes included in the model: While removing 167 sustained electrodes did not impact decoding accuracy, removing 43 right rhythmic electrodes reduced decoding accuracy (Fig 7B). This is best illustrated by the ability to reconstruct a recognizable song from the data of a single patient, with 61 electrodes located on the right STG.

This last result shows the feasibility of this stimulus reconstruction approach in a clinical setting and suggests that future BCI applications should target STG implantation sites in conjunction with functional localization rather than solely relying on a high number of electrodes. We also quantified the impact of the dataset duration on decoding accuracy. We found that 80% of the maximum observed decoding accuracy was achieved in 37 seconds, supporting the feasibility of using predictive modeling approaches in relatively small datasets.

Music perception relied on both hemispheres, with a preference for the right hemisphere. The right hemisphere had a higher proportion of electrodes with significant STRFs, higher STRF prediction accuracies, and a higher impact of ablating right electrode sets (both anatomical and functional) from the decoding models. Left hemisphere electrodes also exhibited significant STRFs and a reduced prediction accuracy when ablated. These results are in accord with prior research, showing that music perception relies on a bilateral network, with a relative right lateralization [23,24,59]. We also found that the spatial distribution of musical information within this network differed between hemispheres, as suggested by the ablation results. Specifically, redundant musical information was distributed between STG, SMC, and IFG in the left hemisphere, whereas unique musical information was concentrated in STG in the right hemisphere. Such spatial distribution is reminiscent of the dual-stream model of speech processing [60].

Importantly, we found a critical role of bilateral STG in representing musical information, in line with prior studies [48,54,61,62]. As observed in other studies, STRFs obtained from the STG had rich, complex tuning patterns. We identified 4 components: onset, sustained, late onset, and rhythmic. The onset and sustained components were similar to those observed for speech in prior work [55,56] and were also observed in anatomically distinct STG subregions, with the onset component in posterior STG and the sustained component in mid- and anterior STG. The onset component was tuned to a broad range of frequencies but to a narrow time window peaking at 90 ms, consistent with the lag at which HFA tracked music intensity profile [24]. This component was

not speech specific as it was activated by both vocals and instrumental onsets, consistent with prior speech work [56]. The sustained component, however, was only activated by vocals. The late onset component was found in electrodes neighboring the onset component in STG and had similar tuning properties as the onset component, only peaking at a later latency of 210 ms. This is in line with the findings of Nourski and colleagues [63], who, using click trains and a speech syllable, observed a concentric spatial gradient of HFA onset latencies in STG, with shorter latencies in post-/mid-STG and longer latencies in surrounding tissue. We also observed a rhythmic component located in mid-STG, which was tuned to the 6.66-Hz 16th notes of the rhythm guitar. This uncovers a novel link between HFA and a specific rhythmic signature in a subregion of STG, expanding prior studies that found an involvement of STG in a range of rhythmic processes [64–66]. Together, these 4 components paint a rich picture of the anatomo-functional organization of complex sound processing in the human STG.

Future research could target extending electrode coverage to additional regions, varying the models' features and targets, or adding a behavioral dimension. Note we lacked coverage in the primary auditory cortex (A1), which could have improved performance of the linear decoding models. Importantly, the encoding models we used in this study to investigate the neural dynamics of music perception estimated the linear relationship between song's acoustics and elicited HFA. It is possible that regions not highlighted by our study respond to the song, either in other neural frequency bands [35,67] or encoding higher-order musical information (e.g., notes, chords, degree of dissonance or of syncopation). Finally, we lacked patient-related information about musicianship status or degree of familiarity with the song, preventing us from investigating interindividual variability.

Combining unique iEEG data and modeling-based analyses, we provided the first recognizable song reconstructed from direct brain recordings. We showed the feasibility of applying predictive modeling on a relatively short dataset, in a single patient, and quantified the impact of different methodological factors on the prediction accuracy of decoding models. Our results confirm and extend past findings on music perception, including a right-hemisphere preference and a major role of bilateral STG. In addition, we found that the STG encodes the song's acoustics through partially overlapping neural populations tuned to distinct musical elements and delineated a novel STG subregion tuned to musical rhythm. The anatomo-functional organization reported in this study may have clinical implications for patients with auditory processing disorders. For example, the musical perception findings could contribute to development of a general auditory decoder that includes the prosodic elements of speech based on relatively few, well-located electrodes.

We limited our investigation to the auditory spectrogram on the stimulus side and to HFA on the neural activity side, given the complex study design encompassing several higher-order analyses building upon encoding and decoding models. Future studies should explore different higher-order representations of musical information in the auditory brain (i.e., notes, chords, sheet music), as well as lower neural oscillatory bands and spectral components (e.g., theta, alpha, and beta power, aperiodic component), known to represent relevant acoustic information, adding another brick in the wall of our understanding of music processing in the human brain.

## Methods

### Ethics statement

All patients volunteered and gave their written informed consent prior to participating in the study. The experimental protocol has been approved in accordance with the Declaration of Helsinki by the Institutional Review Boards of both the Albany Medical College (IRB #2061) and the University of California, Berkeley (CPHS Protocol #2010-01-520).

## Participants

Twenty-nine patients with pharmacoresistant epilepsy participated in the study (15 females; age range 16 to 60, mean 33.4 ± SD 12.7; 23 right-handed; full-scale intelligence quotient range 74 to 122, mean 96.6 ± SD 13.1). All had intracranial grids or strips of electrodes (ECoG) surgically implanted to localize their epileptic foci, and electrode location was solely guided by clinical concern. Recordings took place at the Albany Medical Center (Albany, NY). All patients had self-declared normal hearing.

## Task

Patients passively listened to the song *Another Brick in the Wall*, *Part 1*, by *Pink Floyd* (released on the album The Wall, Harvest Records/Columbia Records, 1979). They were instructed to listen attentively to the music, without focusing on any special detail. Total song duration was 190.72 seconds (waveform is represented in Fig 1A, top). The auditory stimulus was digitized at 44.1 kHz and delivered through in-ear monitor headphones (bandwidth 12 Hz to 23.5 kHz, 20 dB isolation from surrounding noise) at a comfortable sound level adjusted for each patient (50 to 60 dB SL). Eight patients had more than one recording of the present task, in which cases we selected the cleanest one (i.e., containing the least epileptic activity or noisy electrodes).

## Intracranial recordings

Direct cortical recordings were obtained through grids or strips of platinum-iridium electrodes (Ad-Tech Medical, Oak Creek, WI), with center-to-center distances of 10 mm for 21 patients, 6 mm for four, 4 mm for three, or 3 mm for one. We recruited patients in the study if their implantation map covered at least partially the STG (left or right). The cohort consists of 28 unilateral cases (18 left, 10 right) and 1 bilateral case. Total number of electrodes across all 29 patients was 2,668 (range 36 to 250, mean 92 electrodes). ECoG activity was recorded at a sampling rate of 1,200 Hz using g.USBamp biosignal acquisition devices (g.tec, Graz, Austria) and BCI2000 [68].

## Preprocessing—Auditory stimulus

To study the relationship between the acoustics of the auditory stimulus and the ECoG-recorded neural activity, the song waveform was transformed into a magnitude-only auditory spectrogram using the NSL MATLAB Toolbox [69]. This transformation mimics the processing steps of early stages of the auditory pathways, from the cochlea's spectral filter bank to the midbrain's reduced upper limit of phase-locking ability, and outputs a psychoacoustic-, neurophysiologic-based spectrotemporal representation of the song (similar to the cochleagram) [70,71]. The resulting auditory spectrogram has 128 frequency bins from 180 to 7,246 Hz, with characteristic frequencies uniformly distributed along a logarithmic frequency axis (24 channels per octave), and a sampling rate of 100 Hz. This full-resolution, 128-frequency bin spectrogram is used in the song reconstruction analysis. For all other analyses, to decrease the computational load and the number of features, we output a reduced spectrogram with 32 frequency bins from 188 to 6,745 Hz (Fig 1A, bottom).

## Preprocessing—ECoG data

We used the HFA (70 to 150 Hz) as an estimate of local neural activity [72] (Fig 1C). For each dataset, we visually inspected raw recorded signals and removed electrodes exhibiting noisy or epileptic activity, with the help of a neurologist (RTK). Overall, from our starting set of 2,668

electrodes, we removed 106 noisy electrodes (absolute range 0 to 22, mean 3.7 electrodes; relative range 0% to 20.2%, mean 3.7%) and 183 epileptic electrodes (absolute range 0 to 28, mean 6.3; relative range 0% to 27.6%, mean 7.6%) and obtained a set of 2,379 artifact-free electrodes.

We then extracted data aligned with the song stimulus, adding 10 seconds of data padding before and after the song (to prevent filtering-induced edge artifacts). We filtered out power-line noise, using a range of notch filters centered at 60 Hz and harmonics up to 300 Hz (Butterworth, fourth order, 2 Hz bandwidth), and removed slow drifts with a 1-Hz high-pass filter (Butterworth, fourth order). We used a bandpass–Hilbert approach [73] to extract HFA, with 20-Hz-wide sub-bands spanning from 70 to 150 Hz in 5-Hz steps (70 to 90, 75 to 95, . . . up to 130 to 150 Hz). We chose a 20-Hz bandwidth to enable the observation of temporal modulations up to 10 Hz [74], encompassing the 6.66-Hz 16th-note rhythm guitar pattern, pervasive throughout the song. This constitutes a crucial methodological point, enabling the observation of the rhythmic component (Fig 3D). For each sub-band, we first bandpass-filtered the signal (Butterworth, fourth order), then performed median-based common average reference (CAR) [75], and computed the Hilbert transform to obtain the envelope. We standardized each sub-band envelope using robust scaling on the whole time period (subtracting the median and dividing by the interquartile range between the 10th and 90th percentiles) and averaged them together to yield the HFA estimate. We performed CAR separately for electrodes plugged on different splitter boxes to optimize denoising in 14 participants. Finally, we removed the 10-second pads, down-sampled data to 100 Hz to match the stimulus spectrogram's sampling rate, and tagged outlier time samples exceeding 7 standard deviations for later removal in the modeling preprocessing. We used FieldTrip [76] (version from May 11, 2021) and homemade scripts to perform all the above preprocessing steps. Unless specified otherwise, all further analyses and computations were implemented in MATLAB (The MathWorks, Natick, MA, USA; version 2021a).

### Preprocessing—Anatomical data

We followed the anatomical data processing pipeline presented in Stolk and colleagues [77] to localize electrodes from a preimplantation MRI, a postimplantation CT scan, and coverage information mapping electrodes to channel numbers in the functional data. After coregistration of the CT scan to the MRI, we performed brain-shift compensation with a hull obtained using scripts from the iso2mesh toolbox [78,79]. Cortical surfaces were extracted using the FreeSurfer toolbox [80]. We used volume-based normalization to convert patient-space electrode coordinates into MNI coordinates for illustration purposes, and surface-based normalization using the FreeSurfer's fsaverage template to automatically obtain anatomical labels from the aparc+aseg atlas. Labels were then confirmed by a neurologist (RTK).

### Encoding—Data preparation

We used STRFs as encoding models, with the 32 frequency bins of the stimulus spectrogram as features or predictors, and the HFA of a given electrode as target to be predicted.

We log-transformed the auditory spectrogram to compress all acoustic features into the same order of magnitude (e.g., low-sound-level musical background and high-sound-level lyrics). This ensured modeling would not be dominated by high-volume musical elements.

We then computed the feature lag matrix from the song's auditory spectrogram (Fig 1D). As HFA is elicited by the song stimulus, we aim at predicting HFA from the preceding song spectrogram. We chose a time window between 750 ms and 0 ms before HFA to allow a sufficient temporal integration of auditory-related neural responses, while ensuring a reasonable features-to-observations ratio to avoid overfitting. This resulted in 2,400 features (32 frequency bins by 75 time lags at a sampling rate of 100 Hz).

We obtained 18,898 observations per electrode, each one consisting of a set of 1 target HFA value and its preceding 750-ms auditory spectrogram excerpt (19,072 samples of the whole song, minus 74 samples at the beginning for which there is no preceding 750-ms window).

At each electrode, we rejected observations for which the HFA value exceeded 7 standard deviations (Z units), resulting in an average rejection rate of 1.83% (min 0%—max 15.02%, SD 3.2%).

## Encoding—Model fitting

To obtain a fitted STRF for a given electrode, we iterated through the following steps 250 times.

We first split the dataset into training, validation, and test sets (60–20–20 ratio, respectively) using a custom group-stratified-shuffle-split algorithm (based on the StratifiedShuffleSplit cross-validator in scikit-learn). We defined relatively long, 2-second groups of consecutive samples as indivisible blocks of data. This ensured that training and test sets would not contain neighboring, virtually identical samples (as both music and neural data are highly correlated over short periods of time) and was critical to prevent overfitting. We used stratification to enforce equal splitting ratios between the vocal (13 to 80 seconds) and instrumental parts of the song. This ensured stability of model performance across all 250 iterations, by avoiding that a model could be trained on the instrumentals only and tested on the vocals. We used shuffle splitting, akin to bootstrapping with replacement between iterations, which allows us to determine test set size independently from the number of iterations (as opposed to KFold cross-validation).

We then standardized the features by fitting a robust scaler to the training set only (estimates the median and the 2 to 98 quantile range; RobustScaler in sklearn package) and using it to transform all training, validation, and test sets. This gives comparable importance to all features, i.e., every time lag and frequency of the auditory spectrogram.

We employed linear regression with RMSProp optimizer for efficient model convergence, Huber loss cost function for robustness to outlier samples, and early stopping to further prevent overfitting. In early stopping, a generalization error is estimated on the validation set at each training step, and model fitting ends after this error stops diminishing for 10 consecutive steps. This model was implemented in Tensorflow 1.6 and Python 3.6. The learning rate hyperparameter of the RMSProp optimizer was manually tuned to ensure fast model convergence while also avoiding exploding gradients (overshooting of the optimization minimum).

We evaluated prediction accuracy of the fitted model by computing both the correlation coefficient (Pearson's r) and the r-squared between predicted and actual test set target (i.e., HFA at a given electrode). Along with these 2 performance metrics, we also saved the fitted model coefficients.

Then, we combined these 250 split-scale-fit-evaluate iterations in a bootstrap-like approach to obtain 1 STRF and assess its significance (i.e., whether we can linearly predict HFA, at a given electrode, from the song spectrogram). For each STRF, we z-scored each coefficient across the 250 models (Fig 1E). For the prediction accuracy, we computed the 95% confidence interval (CI) from the 250 correlation coefficients and deemed an electrode as significant if its 95% CI did not contain 0. As an additional criterion, we rejected significant electrodes with an average r-squared (across the 250 models) at or below 0.

## Encoding—Analysis of prediction accuracy

To assess how strongly each brain region encodes the song, we performed a two-way ANOVA on the correlation coefficients of all electrodes showing a significant STRF, with laterality (left

or right hemisphere) and area (STG, sensorimotor, IFG, or other) as factors. We then performed a multiple comparison (post hoc) test to disentangle any differences between factor levels.

### Decoding—Parametric analyses

We quantified the influence of different methodological factors (number of electrodes, dataset duration, and model type) on the prediction accuracy of decoding models. In a bootstrapping approach, we randomly constituted subsets of 5, 10, 20, 40, 80, 160, and 320 electrodes (sampling without replacement), regardless of their anatomical location, to be used as inputs of linear decoding models. We processed 100 bootstrap resamples (i.e., 100 sets of 5 electrodes, 100 sets of 10 electrodes. . .) and normalized for each of the 32 frequency bins the resulting correlation coefficients by the correlation coefficients of the full, 347-electrode decoding model. For each resample, we averaged the correlation coefficients from all 32 models (1 per frequency bin of the song spectrogram). This yielded 100 prediction accuracy estimates per number of electrodes. We then fitted a two-term power series model to these estimates to quantify the apparent power-law behavior of the obtained bootstrap curve. We adopted the same approach for dataset duration, with excerpts of 15, 30, 60, 90, 120, 150, and 180 consecutive seconds.

To investigate the impact of model type on decoding accuracy and to assess the extent to which we could reconstruct a recognizable song, we trained linear and nonlinear models to decode each of the 128 frequency bins of the full spectral resolution song spectrogram from HFA of all 347 significant electrodes. We used the MLP—a simple, fully connected neural network—as a nonlinear model (MLPRegressor in sklearn). We chose an MLP architecture of 2 hidden layers of 64 units each, based both on an extension of the *Universal Approximation Theorem* stating that a 2 hidden layer MLP can approximate any continuous multivariate function [38] and on a previous study with a similar use case [35]. Since MLP layers are fully connected (i.e., each unit of a layer is connected to all units of the next layer), the number of coefficients to be fitted is drastically increased relatively to linear models (in this case, $F * N + N * N + N$ vs. $F$, respectively, where the total number of features $F = E * L$, with E representing the number of significant electrodes included as inputs of the decoding model, and L the number of time lags, and N represents the number of units per layer). Given the limited dataset duration, we reduced time lags to 500 ms based on the absence of significant activity beyond this point in the STRF components and used this L value in both linear and nonlinear models.

We defined a fixed, 15-second continuous test set during which the song contained both vocals and instrumentals (from 61 to 76 seconds of the original song, accessible on any streaming service) and held it out during hyperparameter tuning and model fitting. We tuned model hyperparameters (learning rate for linear models, and L2-regularization alpha for MLPs) through 10-resample cross-validation. We performed a grid search on each resample (i.e., training/validation split) and saved for each resample the index of the hyperparameter value yielding the minimum validation mean squared error (MSE). Candidate hyperparameter values ranged between .001 and 100 for the learning rate of linear models, and between .01 and 100 for the alpha of MLPs. We then rounded the mean of the 10 resulting indices to obtain the cross-validated, tuned hyperparameter. As a homogeneous presence of vocals across training, validation, and test sets was crucial for proper tuning of the alpha hyperparameter of MLPs, we increased group size to 5 seconds, equivalent to about 2 musical bars, in the group-stratified-shuffle-split step (see Encoding—Model fitting for a reference), and used this value for both linear and nonlinear models. For MLPs specifically, as random initialization of coefficients could lead to convergence towards local optima, we adopted a best-of-3 strategy where we only kept the "winning" model (i.e., yielding the minimum validation MSE) among 3 models fitted on the same resample.

Once we obtained the tuned hyperparameter, we computed 100 models on distinct training/ validation splits, also adopting the best-of-3 strategy for the nonlinear models (this time keeping the model yielding the maximum test r-squared). We then sorted models by increasing r-squared and evaluated the "effective" r-squared by computing the r-squared between the test set target (the actual amplitude time course of the song's auditory spectrogram frequency bin) and averages of n models, with n varying from 100 to 1 (i.e., effective r-squared for the average of all 100 models, for the average of the 99 best, . . ., of the 2 best, of the best model). Lastly, we selected n based on the value giving the best effective r-squared and obtained a predicted target along with its effective r-squared as an estimate of decoding accuracy. The steps above were performed for all 128 frequency bins of the song spectrogram, both for linear and nonlinear models, and we compared the resulting effective r-squared using a two-sided paired *t* test.

## Decoding—Song waveform reconstruction

To explore the extent to which we could reconstruct the song from neural activity, we collected the 128 predicted targets for both linear and MLP decoding models as computed above, therefore assembling the decoded auditory spectrograms. To denoise and improve sound quality, we raised all spectrogram samples to the power of 2, thus highlighting prominent musical elements such as vocals or lead guitar chords, relative to background noise. As both magnitude and phase information are required to reconstruct a waveform from a spectrogram, we used an iterative phase-estimation algorithm to transform the magnitude-only decoded auditory spectrogram into the song waveform (*aud2wav*) [69]. To have a fair basis against which we could compare the song reconstruction of the linearly and nonlinearly decoded spectrograms, we transformed the original song excerpt corresponding to the fixed test set into an auditory spectrogram, discarded the phase information, and applied this algorithm to revert the spectrogram into a waveform (S1 Audio). We performed 500 iterations of this aud2wav algorithm, enough to reach a plateau where error did not improve further.

## Decoding—Song-excerpt identification rank analysis

To evaluate the quality of the decoded song spectrogram, we used an objective approach based on correlation [34]. Firstly, we divided the song into twelve 15-second segments. We then decoded each one of these segments as held-out test sets, using both linear and nonlinear models. Next, we divided all predicted 15-second spectrograms into 5-second excerpts. We computed the 2D correlation coefficients between each of the 38 decoded excerpts and all 38 original excerpts. We then performed an excerpt identification rank analysis by sorting these coefficients in ascending order and by identifying the rank of the actual excerpt correlation. For example, for decoded excerpt #4, if the correlation coefficient with the original excerpt #4 is the third best, its rank will be 36/38. The resulting metric ranges from 1/38 (worst possible identification, i.e., the given decoded excerpt matches best with all other song excerpts than with its corresponding one) to 38/38, and averaging ranks across all song excerpts gives a proxy for classification ability of the linear and nonlinear models. To assess statistical significance, we computed 1,000 iterations of the algorithm above while randomly permuting indices of the original song excerpts to obtain a null distribution of the mean normalized rank. We deemed the mean normalized rank of our linear and nonlinear decoding models as significant if they were outside of the 95% CI (i.e., exceeded the 97.5 percentile) of the null distribution.

## Encoding—Analysis of model coefficients

We analyzed the STRF tuning patterns using an ICA to highlight electrode populations tuned to distinct STRF features. Firstly, we ran an ICA with 10 components on the centered STRF

coefficients to identify components individually explaining more than 5% of variance. We computed explained variance by back-projecting each component and using the following formula: $\text{pvaf}_i = 100 - 100 \times \text{mean}(\text{var}(\text{STRF} - \text{backproj}_i)) / \text{mean}(\text{var}(\text{STRF}))$, with i ranging from 1 to 10 components, $\text{pvaf}_i$ being the percentage of variance accounted for by ICA component i, STRF being the centered STRF coefficients, and $\text{backproj}_i$ being the back-projection of ICA component i in electrode space. We found 3 ICA components, each explaining more than 5% of variance, and together explaining 52.5% variance. To optimize the unmixing process, we ran a new ICA asking for 3 components. Then, we determined each component sign by setting as positive the sign of the most salient coefficient (i.e., with the highest absolute value, or magnitude). Lastly, for each ICA component, we defined electrodes as representing the component if their ICA coefficient was positive.

To look at rhythmic tuning patterns, we computed the temporal modulations of each STRF. Indeed, due to their varying frequencies and latencies, they were not captured by the combined component analysis. We quantified temporal modulations between 1 and 16 Hz over the 32 spectral frequency bins of each STRF and extracted the maximum modulation value across all 32 frequency bins between 6 and 7 Hz of temporal modulations, corresponding to the song rhythmicity of 16th notes at 99 bpm. We defined electrodes as representing the component if their maximum modulation value was above a manually defined threshold of .3.

### Encoding—Musical elements

To link STRF components to musical elements in the song, we ran a sliding-window correlation between each component and the song spectrogram. Positive correlation values indicate specific parts of the song or musical elements (i.e., vocals, lead guitar...) that elicit an increase of HFA.

### Decoding—Ablation analysis

To assess the contribution of different brain regions and STRF components in representing the song, we performed an ablation analysis. We quantified the impact of ablating sets of electrodes on the prediction accuracy of a linear decoding model computed using all 347 significant electrodes. Firstly, we constituted sets of electrodes based on anatomical or functional criteria. We defined 12 anatomical sets by combining 2 factors—area (whole hemisphere, STG, SMC, IFG, or other areas) and laterality (bilateral, left or right). We defined 12 functional sets by combining 2 factors—STRF component identified in the STRF coefficient analyses (onset, sustained, late onset, and rhythmic) and laterality (bilateral, left or right). See Fig 5 for the exact list of electrode sets. Secondly, we computed the decoding models using the same algorithm as for the encoding models. Decoding models aim at predicting the song spectrogram from the elicited neural activity. Here, we used HFA from a set of electrodes as input and a given frequency bin of the song spectrogram as output. For each of the 24 ablated sets of electrodes, we obtained 32 models (1 per spectrogram frequency bin) and compared each one of them to the corresponding baseline model computed using all 347 significant electrodes (repeated-measure one-way ANOVA). We then performed a multiple comparison (post hoc) test to assess differences between ablations.

We based our interpretation of ablation results on the following assumptions. Collectively, as they had significant STRFs, all 347 significant electrodes represent acoustic information on the song. If ablating a set of electrodes resulted in a significant impact on decoding accuracy, we considered that this set represented unique information. Indeed, were this information shared with another set of electrodes, a compensation-like mechanism could occur and void the impact on decoding accuracy. If ablating a set of electrodes resulted in no significant

impact on decoding accuracy, we considered that this set represented redundant information, shared with other electrodes (as the STRFs were significant, we ruled out the possibility that it could be because this set did not represent any acoustic information). Also, comparing the impact of a given set and one of its subsets of electrodes provided further insights on the unique or redundant nature of the represented information.

Note that we performed this ablation analysis on linear decoding models to ensure interpretability of the results. Indeed, as deep neural networks are able to model any function [38], they could reconstitute acoustic information (e.g., when ablating STG) from higher-order, nonlinear representation of musical information (e.g., in SMC or IFG) and could thus alleviate, if not mask entirely, any impact on decoding accuracy. Using linear decoding models restricts compensation to the same (or at most, linearly related) information level and enables drawing conclusions from the ablation analysis results. Further, compared to linear models, nonlinear models require tuning more hyperparameters, with most likely different optimal values between ablations, which could bias the results.

### Preventing overfitting

Given the claims of this paper are based on the results of encoding and decoding models, it was crucial to make sure we avoid overfitting. We implemented state-of-the-art practices at all steps. Before splitting, we assessed autocorrelation in both the stimulus and the neural data time series and defined 2-second groups of consecutive samples as indivisible blocks of data to be allocated to either the training, validation, or test set (5-second groups for song reconstruction). For data scaling, we fitted the scaler on the training set and applied it to both the validation and test sets (scaling the whole time series, as sometimes seen in the literature, allows the scaler to learn the statistics of the test set, possibly leading to overfitting). Most importantly, we used early stopping for our encoding and decoding models, which, by definition, stops training as soon as the model starts to overfit and stops generalizing to the validation set, and L2 regularization for our nonlinear models, which constricts model coefficients to prevent overfitting. Finally, we made sure to evaluate all models on held-out test sets.

### Reference gender statistics

Across all 80 references, 7 had females as first and last authors, 9 had a male first author and a female last author, 16 had a female first author and a male last author, and 38 had males as first and last authors. Ten papers had a single author, among which one was written by a female.

### Supporting information

**S1 Fig. Electrode coverage for each patient.** All presented electrodes are free of any artifactual or epileptic activity. Red marker color indicates song-responsive electrodes. For patients with low-resolution MR images, electrode coverage is plotted on the MNI template (asterisk symbols after patient code). Note that patient P27 had bilateral coverage. The data underlying this figure can be obtained at https://doi.org/10.5281/zenodo.7876019.
(TIF)

**S2 Fig. Single-patient linear decoding.** On the y-axis, 100% represents the maximum decoding accuracy, obtained using all 347 significant electrodes across all 29 patients. The black curve shows data points obtained from a 100-resample bootstrapping analysis, while the red curve shows a two-term power series fit line. Error bars indicate SEM. Colored dot markers represent prediction accuracy for single-patient decoding. For example, P1 had 7 significant electrodes used as features in the P1-only decoding models, which reached 43.7% of best

prediction accuracy (dark blue dot). The blue curve shows a two-term power series line fitted on these single-patient prediction accuracy data points. The data underlying this figure can be obtained at https://doi.org/10.5281/zenodo.7876019.
(TIF)

**S3 Fig. Single-patient nonlinear reconstruction.** Auditory spectrograms of the reconstructed song using nonlinear models from electrodes of patient P28 only (top), P15 only (middle), and P16 only (bottom). Corresponding audio waveforms can be listened to in S5, S6, and S7 Audio files, respectively. The data underlying this figure can be obtained at https://doi.org/10.5281/zenodo.7876019.
(TIF)

**S4 Fig. STRFs for the 239 artifact-free electrodes of patient P29.** Red labels represent significant STRFs, and the respective prediction accuracies (Pearson's r) are shown above each significant STRF on the right. Color code for STRF coefficients is identical to the one used in Fig 1. Anatomical axes are plotted in the top right corner (posterior on the left, in this right hemisphere coverage). Anatomical landmarks are shown as acronyms in bold font (ITS, inferior temporal sulcus; ITG, inferior temporal gyrus; LS, lateral sulcus, also called Sylvian fissure; MTG, middle temporal gyrus; STG, superior temporal gyrus; STS, superior temporal sulcus). The data underlying this figure can be obtained at https://doi.org/10.5281/zenodo.7876019.
(TIF)

**S1 Audio. Original song waveform transformed into a magnitude-only auditory spectrogram, then transformed back into a waveform using an iterative phase-estimation algorithm.** As an assessment of the impact of this spectrogram-to-waveform transformation on sound quality, it constitutes a perceptual upper bound for the following HFA-based reconstructions. The data underlying this figure can be obtained at https://doi.org/10.5281/zenodo.7876019.
(WAV)

**S2 Audio. Reconstructed song excerpt using linear models fed with all 347 significant electrodes from all 29 patients.** The data underlying this figure can be obtained at https://doi.org/10.5281/zenodo.7876019.
(WAV)

**S3 Audio. Reconstructed song excerpt using nonlinear models fed with all 347 significant electrodes from all 29 patients.** The data underlying this figure can be obtained at https://doi.org/10.5281/zenodo.7876019.
(WAV)

**S4 Audio. Reconstructed song excerpt using nonlinear models fed with the 61 significant electrodes (3-mm center-to-center distance) from P29.** The data underlying this figure can be obtained at https://doi.org/10.5281/zenodo.7876019.
(WAV)

**S5 Audio. Reconstructed song excerpt using nonlinear models fed with the 23 significant electrodes (10-mm center-to-center distance) from P28.** The data underlying this figure can be obtained at https://doi.org/10.5281/zenodo.7876019.
(WAV)

**S6 Audio. Reconstructed song excerpt using nonlinear models fed with the 17 significant electrodes (6-mm center-to-center distance) from P15.** The data underlying this figure can

be obtained at https://doi.org/10.5281/zenodo.7876019.
(WAV)

**S7 Audio. Reconstructed song excerpt using nonlinear models fed with the 10 significant electrodes (10-mm center-to-center distance) from P16.** The data underlying this figure can be obtained at https://doi.org/10.5281/zenodo.7876019.
(WAV)

## Acknowledgments

We thank Brian N. Pasley, Christopher R. Holdgraf, Stephanie Martin, Christian Mikutta, Randolph F. Helfrich, and Arjen Stolk for their helpful comments on data analysis.

## Author Contributions

**Conceptualization:** Ludovic Bellier, Robert T. Knight.

**Data curation:** Aysegul Gunduz, Gerwin Schalk, Peter Brunner.

**Formal analysis:** Ludovic Bellier.

**Funding acquisition:** Ludovic Bellier, Peter Brunner, Robert T. Knight.

**Investigation:** Aysegul Gunduz, Gerwin Schalk, Peter Brunner.

**Methodology:** Ludovic Bellier.

**Project administration:** Robert T. Knight.

**Resources:** Peter Brunner.

**Software:** Ludovic Bellier.

**Supervision:** Robert T. Knight.

**Visualization:** Ludovic Bellier.

**Writing – original draft:** Ludovic Bellier, Anaïs Llorens, Déborah Marciano.

**Writing – review & editing:** Ludovic Bellier, Anaïs Llorens, Déborah Marciano, Robert T. Knight.

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
