## [Editor Report · Decision Letter 0]

1 Oct 2022

Dear Dr Bellier, 

Thank you for submitting your manuscript entitled "Nonlinear decoding models enable music reconstruction from human auditory cortex activity" for consideration as a Research Article by PLOS Biology.

Your manuscript has now been evaluated by the PLOS Biology editorial staff, and has been briefly discussed with an academic editor with relevant expertise. I am writing to let you know that we would like to send your submission out for external peer review. However, please note that, personal constraints on the part of our academic editor prevented us from having a more detailed discussion about this work. I therefore must stress that we are somewhat one the fence as to whether the overall advance of this work would be suitable for PLOS Biology or not and we will be looking for enthusiasm from the reviewers on this issue.

Before we can send your manuscript to reviewers, we need you to complete your submission by providing the metadata that is required for full assessment. To this end, please login to Editorial Manager where you will find the paper in the 'Submissions Needing Revisions' folder on your homepage. Please click 'Revise Submission' from the Action Links and complete all additional questions in the submission questionnaire.

Once your full submission is complete, your paper will undergo a series of checks in preparation for peer review. After your manuscript has passed the checks it will be sent out for review. To provide the metadata for your submission, please Login to Editorial Manager (https://www.editorialmanager.com/pbiology) within two working days, i.e. by Oct 03 2022 11:59PM.

Kind regards,

Kris

Kris Dickson, Ph.D. (she/her)

Neurosciences Senior Editor/Section Manager

PLOS Biology

kdickson@plos.org

---

## [Decision Letter · Decision Letter 1]

21 Nov 2022

Dear Dr Bellier,

Thank you for your patience while your manuscript "Nonlinear decoding models enable music reconstruction from human auditory cortex activity" was peer-reviewed at PLOS Biology. It has now been evaluated by the PLOS Biology editors, an Academic Editor with relevant expertise, and by several independent reviewers. 

Based on the reviewers' feedback, we are happy to have you revise this work to address the reviewer concerns. When doing so, and given the broad scope of our journal, we'd also ask you to more explicitly discuss how these findings contribute to our broader understanding of human auditory processing mechanisms.

Please note that we cannot make any final decision about publication until we have seen the revised manuscript and your response to the reviewers' comments. Your revised manuscript is likely to be sent for further evaluation by all or a subset of the reviewers.

**IMPORTANT - SUBMITTING YOUR REVISION**

*Re-submission Checklist*

*Published Peer Review*

*PLOS Data Policy*

*Blot and Gel Data Policy*

Sincerely,

Kris

Kris Dickson, Ph.D., (she/her)

Neurosciences Senior Editor/Section Manager

PLOS Biology

kdickson@plos.org

REVIEWS:

Do you want your identity to be public for this peer review?

Reviewer #1: Yes: Marco Tettamanti

Reviewer #2: No

Reviewer #3: No

Reviewer #1: Bellier and colleagues analyzed a previously collected iEEG dataset in 29 pharmacoresistant epileptic patients who were implanted with ECoG electrodes prior to neurosurgery, with high-frequency neural activity measured while passively listening to a popular Pink Floyd's song. Using combined regression-based decoding models and encoding analyses, they successfully reconstructed multidimensional, acoustic song features from recorded neural activity. These results are both original and important, as they reveal a) information processing content in the STG during music perception, showing the feasibility of reconstructing perceived music from high-frequency neural activity, along with b) a new STG sub-region tuned to musical rhythm encoding.

While previous published studies have applied decoding models to decode music-induced neural activity with a classification approach, the present study is the first to show the reconstruction of music through regression-based decoding models, by analogy to previous work with speech stimuli (AnumanchipalliG et al. Nature 2019).

The manuscript is clearly written and the methodology and research questions are sound. However, several issues require further clarification.

Essential points:

1. Fig. Supp. 1 shows that some patients had vastly more song-responsive electrodes than others (e.g., P22, P23, P29). In fact, in Fig. Supp. 2 it is quite clear that some patients have rather low decoding accuracy. How does the number of song-responsive electrodes affect decoding accuracy in each patient? Would removing data of any patients with a low number of song-responsive electrodes and/or low decoding accuracy leave the overall decoding results unaffected?

The authors make a case on the feasibility of predictive modeling even in individual patients, but since they show it only in the patient with the most song-responsive electrodes in the right hemisphere (P29), it is not clear to what extent the results of the present study are representative of only a subset of patients or the group in general.

2. Methods, pp.29-31: encoding analysis. The authors made a number of decisions on parameter values in order to prevent model overfitting. While this reflects good practices, it does not guarantee that the problem of overfitting will be eliminated entirely. Can the authors provide an estimate on the degree of protection against overfitting in their model? Have they tested it against a range of different parameter values?

3. Results, p.20: "Furthermore, the fact that removing single regions in the left hemisphere had no impact but removing all left electrodes did, suggests redundancy within the left hemisphere, with musical information being spatially

distributed across left hemisphere regions."

Could it be in turn that there is a mass effect, such that removing all left electrodes leads to a sharp drop in decoding accuracy due to insufficient remaining patient/electrode data? This may be the case because there were overall more left than right unilateral patients and overall more song-responsive electrodes in the left hemisphere than in the right hemisphere.

On a related question, with respect to Fig.3A, it would be important to know if the anatomical location of the electrodes that provide best prediction accuracy matters (for example, where are the 43 electrodes that contribute to the 80% best prediction accuracy concentrated in the bootstrap analysis?).

Other points:

4. Introduction, p.3: The intricate issue of overlap of brain structures and hierarchical information structure between music and speech is dismissed somewhat superficially, with reference only to a couple of relatively older citations. A more up-to-date and balanced view should be provided.

5. Methods: p.26: "Eight patients had more than one recording of the present task, in which cases we selected the cleanest one (i.e., containing the least epileptic activity or noisy electrodes)."

Information on frequency and distribution of epileptic activity in each patient's data should be provided, along with proportions of data deletion.

6. Results, p.7: the higher proportion of song-responsive electrodes in the right hemisphere is presented as significant in a chi-square test. However, from inspection of Fig.2A it appears that this result could be biased by the higher concentration of left-hemispheric electrodes in regions that are minimally responsive to song stimuli, such as the left temporal pole and left DLPFC. What if this analysis were restricted to regions of interest, like auditory cortices? Despite the limited contribution of this finding to the overall picture, some rephrasing is needed with respect to this point.

7. Decoding Ablation analysis: why did the authors use a linear rather than a non-linear decoding model for the ablation analysis?

-------------------

Reviewer #2: I'm glad the authors pursued this study. I have been wondering about this.

This manuscript is a good contribution to the literature.

I have a couple of minor nits.

1) It might be worth stating up front that this paper is reconstructing a full spectrogram, not the envelope (and other features) that are usually done for (EEG) speech decoding.

2) Why isn't ridge regression directly used for the decoding? This seems more reliable than a method based on SGD.

3) For comparison purposes it would be good to mention the decoding accuracy (either envelope or spectrograms) for pure speech data. Do the authors have that from previous studies?

It is nice to see the ICA analysis of STRFs, and the resulting distributions of types.

Good job.

-----------------

Reviewer #3: Bellier et al. carry out an interesting study centred around the reconstruction of music stimuli form human electrophysiology data. The manuscript is well written, the analyses are rigorous and well described. I only have a few of minor remarks.

L[ine] number: [original text] comment/proposed edit

L27: [recognizable song] the term "recognition" is used throughout the manuscript to indicate that song segments reconstructed from brain activity were more strongly correlated (in the spectral domain) with the audio signal of the same song segment than with audio signals from different song segments. This correlation approach is legitimate, and indeed addresses in some form the confusability of the reconstructions with different segments of the same song. However, I wonder whether "recognizable song" might be completely appropriate, as it implies an experiment assessing recognition with human listeners, experiment that was not carried out by the authors.

L47: [approaches] approach

L88: [While investigating to what extent the song stimulus could be reconstructed from the elicited HFA using a regression approach] why did you focus on HFA only? This needs to be justified, as low-frequency brain signals can indeed contain a lot of information about time-varying acoustic structure.

L89: [three factors on reconstruction accuracy: (1) model type (linear versus nonlinear), (2) electrode density (the number of electrodes used as inputs in decoding models), and (3) dataset duration.] This appears more like a manipulation conceived to assess the performance of the reconstruction method rather than a manipulation designed to answer specific questions about the function of the auditory system.

L112: [Here we investigated whether we could observe similar HFA activity profiles, namely onset and sustained, in response to a musical stimulus.] Why is this particular analysis important to our knowledge of the auditory system?

L118: [and that we would observe some degree of STG parcellation with similar structure as observed in the speech domain.] Can you be more specific? "Some degree of STG parcellation" is perhaps too undefined.

L162: [Analysis of STRF prediction accuracies (Pearson's r) found a main effect of laterality (two-way ANOVA] Specify whether the two-way ANOVA consider simultaneously the effects of cortical region and hemisphere. Was the interaction term modelled? If not, why?

L175: [over 250 resamples] resamples of what?

L185: [100-resample bo¬otstrapping analysis] resampling of what?

L266: [explaining more than 5% variance] EACH explaining more than 5% variance. Also, if I read the labels on panel B well, it might be better to indicate that the 3 ICs explained, together, more than 50% of the variance of the significant STRFs (same comment for line 260).

Fig. 5, panels C and E: the coefficient maps are hard to parse. Would it be possible to show the positive coefficient sensors on top of the negative (black) coefficients? Alternatively, you could show only the positive coefficient sensors here, and put in supplementary materials a figure where you show all sensors.

Fig. 6, panel C: the correspondence between the time-varying correlations and the musical structure (vocals, lead guitar, synth) is hard to parse, as one needs to project mentally the coloured horizontal bars in panel A onto each of the time-varying correlation plots. Would it be possible to replicate the coloured bars in panel A on top of each of the three correlation plots?

L311, L323: [referred from] referred to from

L384: [music perception] music representation or processing would perhaps be more appropriate. Same for other instances.

L504: [magnitude-only auditory spectrogram] specify "(cochleagram)" after spectrogram. Alternatively, use cochleagram throughout the manuscript.

L705: [with i from] with i ranging from

L710: [most salient coefficient] meaning unclear

---

## [Editor Report · Decision Letter 2]

19 May 2023

Dear Dr Bellier,

Thank you for your patience while we considered your revised manuscript "Nonlinear decoding models enable music reconstruction from human auditory cortex activity" for publication as a Research Article at PLOS Biology. This revised version of your manuscript has been evaluated by the PLOS Biology editors and by the Academic Editor who is satisfied by the changes made. 

Based on our Academic Editor's assessment of your revision, we are likely to accept this manuscript for publication, provided you satisfactorily address the remaining editorial requests, in a revision that we think will not take very long. 

IMPORTANT: Please address the following editorial requests: 

1) ETHICS STATEMENT: 

 - Please indicate whether the consent of participants was obtained in a written format.

 - Please provide the approval number(s) of the protocol, approved by the IRB of Albany and UC Berkeley. 

2) DATA REQUEST: Thank you for providing the underlying data for your study as a deposition on Zenodo. Can you please reference this dataset in all relevant figure legends (including supplemental)? For example, you can add the sentence "the data underlying this figure can be obtained at http://doi.org/10.5281/zenodo.7876019"

3) TITLE: We would suggest the title be reorganized to put the finding first. If you agree, we suggest it be changed to something like "Music can be reconstructed from human auditory cortex activity using nonlinear decoding models"

4) Please define BCI in the abstract and manuscript

We expect to receive your revised manuscript within two weeks. 

*Published Peer Review History*

*Press*

Sincerely,

Luke

Lucas Smith, Ph.D.

Associate Editor,

lsmith@plos.org,

PLOS Biology

---

## [Editor Report · Decision Letter 3]

30 May 2023

Dear Dr Bellier,

Thank you for the submission of your revised Research Article "Music can be reconstructed from human auditory cortex activity using nonlinear decoding models" for publication in PLOS Biology and for addressing our last editorial requests in your most recent revision. On behalf of my colleagues and the Academic Editor, David Poeppel, I am pleased to say that we can in principle accept your manuscript for publication, provided you address any remaining formatting and reporting issues. These will be detailed in an email you should receive within 2-3 business days from our colleagues in the journal operations team; no action is required from you until then. Please note that we will not be able to formally accept your manuscript and schedule it for publication until you have completed any requested changes.

PRESS

Sincerely, 

Lucas Smith, Ph.D.,

Associate Editor

PLOS Biology

lsmith@plos.org